# Ex-military personnel's experiences of loneliness and social isolation from discharge, through transition, to the present day

**Suzanne Guthrie-Gower, Gemma Wilson-Menzfeld** *

Faculty of Health and Life Sciences, Northumbria University, Newcastle-upon-Tyne, United Kingdom

* Gemma.Wilson-Menzfeld@northumbria.ac.uk

## Abstract

### Objectives

This study aimed to examine the unique factors of loneliness and social isolation within the ex-military population from discharge, through transition, to the present day.

### Design

A qualitative, Phenomenological approach was adopted.

### Methods

In-depth semi-structured interviews were carried out with 11 participants who had all served in the British Armed Forces and represented all three military services (Royal Navy; Army; Royal Air Force). Reflexive Thematic Analysis was used to analyse the data.

### Results

Three themes were generated—a sense of loss; difficulty in connecting in civilian life; and seeking out familiarity. The findings of this study were examined through the lenses of the Social Needs Approach and the Cognitive Discrepancy Model.

### Conclusions

Individuals developed close bonds in the military through meaningful and prolonged contact, reducing feelings of loneliness and social isolation during their time in service. The sense of belonging was key to social connection, but transition out of the military severed existing relationships, and a lack of belonging hindered the development of relationships within the civilian community. This study has implications for service provision relating to ex-military personnel and future service leavers.

**Data Availability Statement:** Data cannot be shared publicly as it is sensitive qualitative data which would compromise confidentiality if shared.

Sharing this data would breech ethical approval guidelines provided by Northumbria University's ethical approval system (REF: 24393). Please see non-author contact information for the body placing restrictions on this data: (dp.officer@northumbria.ac.uk). Please see non-author contact details for the body imposing restrictions on the data: dp.officer@northumbria.ac.uk.

**Funding:** The author(s) received no specific funding for this work.

**Competing interests:** The authors have declared that no competing interests exist.

## Introduction

Loneliness and social isolation have become increasingly acknowledged as a health concern, are widely recognised risk factors for adverse mental and physical health outcomes [1], and are linked to premature death [2, 3]. This evidence has grown since the COVID-19 pandemic [4] and has highlighted the universal issue of both loneliness and social isolation across the population [5]. Whilst both loneliness and social isolation can be experienced by everyone, the ex-military population present unique experiences of loneliness and social isolation that requires specific attention [6]. Both experiences of loneliness and social isolation in the ex-military population have been associated with post-traumatic stress [7], depression [8], suicidal ideation [9, 10], and deteriorating physical health [11].

The military is a highly structured environment where healthcare, housing, welfare, and social support are intrinsic to the service [12]. In addition to this, the military has distinct social norms that are reinforced during service [12]. Transitioning into a fluid civilian society can, therefore, be challenging without the supportive framework and social norms of the military [13]. Whilst studies on transition from military to civilian life have identified that being socially connected in the community is associated with positive transition [14], there are a number of factors that may be experienced during transition that can impact loneliness and social isolation. The search for continuity from military to civilian life makes adjustment to civilian life challenging [15] which may result in "reverse culture shock" [16] and identity conflict [17, 18].

Comradeship has been found to be a significant factor during military service [19] and a report by the Royal British Legion [20] states that 65% of ex-military personnel exiting the British Armed Forces felt lonely and/or socially isolated. Reduced support networks, as a result of leaving the military, are associated with an increase in loneliness [21] and social isolation increases the propensity to drink to excess at home or alone [22]. Studies have also demonstrated feelings of detachment from both military and civilian society [23, 24] and social ties can weaken [25]. Research from the USA has also identified that ex-military personnel may struggle to build up new social connections when relocating [26], may have difficulty accessing social support to assist with reintegration [27], and may have challenges in retaining or building social networks that support health and wellbeing [28]. One theoretical and methodological perspective around "culturally meaningful networks" examines the social mechanisms which impact social networks through the transition to civilian life [29]. This framework proposes that structure, meaning, and time play crucial roles in influencing social networks after military service [29]. It can therefore be argued that specialised interventions within this subpopulation should consider these three factors, and are essential during both the transitional period and in the longer term [30].

Loneliness has been considered theoretically through both the Social Needs Approach [31] and the Cognitive Discrepancy Model [32]. The Social Needs Approach [31] identifies two distinct categories of loneliness: emotional loneliness, i.e. the absence of an intimate attachment in which individuals may be motivated to seek fulfilling relationships to alleviate their sense of loss; and social loneliness, i.e. a lack of social networks in which individuals may strive to identify new social connections. The Cognitive Discrepancy Model [32] considers loneliness as the perceived discrepancy between the desired and actual quality or quantity of relationships. The model distinguishes between subjective loneliness and objective social isolation and states they may be experienced independently or simultaneously [32].

Developing the narrative around loneliness and social isolation in the British Armed Forces ex-military has been recommended [30] as well as determining the prevalence of both concepts within this subpopulation [6]. However, there is a distinct lack of Social Needs Approach

theoretical application in the wider literature, including literature focusing on experiences of loneliness and social isolation in the Armed Forces Community. Furthermore, conducting qualitative research has been suggested to establish what works for whom and why during transition to civilian life [33] and to provide a deeper understanding of how problems emerge and what contributes to successful outcomes [34]. Given the evidence, this study aimed to examine the unique factors of loneliness and social isolation for ex-military population from discharge, through transition to the present day.

## Materials and methods

### Design

A Phenomenological methodology was adopted as this qualitative method focuses on participants' views of their lived experiences. The phenomenological approach seeks to capture experiences, thoughts and feelings whereby the researcher assumes a person-centred role by listening empathically without questioning or judgement [35]. Grounded theory was not deemed appropriate as it is concerned with the researcher developing a theory from their own interpretations and discourse analysis was not considered suitable as it examines the use of language [35]. Semi-structured interviews were appropriate as they complement the realist phenomenological approach by allowing participants the flexibility to openly describe aspects of loneliness and social isolation that have meaning and relevance to them. Inductive analysis was utilised to analyse the data to ensure findings were data driven rather than seeking out data that upheld a specific theory which is consistent with the phenomenological approach [36]. This study received full ethical approval from Northumbria University's Ethics approval system.

### Participants

Using voluntary and snowballing sampling strategies, participants who were over the age of 18 and had served in the British Armed Forces were recruited (Table 1).

Recruitment was carried out using professional networks known to the researcher (SGG). This network forwarded this email to their own networks and advertised the study via social media. Eleven participants volunteered to participate. Participants were between the age of 49 and 72 (mean = 58) were recruited, six male and five female, all of whom had served in the British Armed Forces. Participants served in the Royal Navy (n = 1), Army (n = 8), and Royal Air Force (n = 2). Length of service ranged from five years to 38 years (mean = 19 years). The

Table 1. Summary of participants.

| Participant number | Sex | Age | Length of Service (years) | Years since discharge |
|---|---|---|---|---|
| Participant 01 | M | 57 | 38 | 3 |
| Participant 02 | M | 55 | 5 | 34 |
| Participant 03 | M | 62 | 20 | 23 |
| Participant 04 | F | 41 | 7 | 15 |
| Participant 05 | F | 66 | 30 | 16 |
| Participant 06 | M | 63 | 28 | 18 |
| Participant 07 | M | 69 | 16 | 30 |
| Participant 08 | F | 72 | 25 | 27 |
| Participant 09 | F | 49 | 13 | 15 |
| Participant 10 | M | 53 | 23 | 13 |
| Participant 11 | F | 49 | 7 | 21 |

number of years elapsed since discharge ranged from three years to 34 years (mean = 20 years) and rank ranged from Private to Lieutenant Colonel.

## Materials

An interview guide was developed consisting of open-ended questions (Fig 1). The interview guide was reviewed by an independent individual, who was ex-military, to establish whether the questions were relevant and appropriate. Following a review by the ethics panel, suggestions were made, and the interview guide was amended accordingly.

## Procedure

An email was distributed amongst a professional network which contained details of the study and the researcher's contact details. Potential participants contacted the researcher who

| Semi-structured interview questions |
|---|
| 1. What were your experiences of comradeship in the military? |
| 2. How did you feel about leaving this behind when you left the military? |
| 3. If you moved to a different area how did this impact you socially? |
| 4. How did you keep connected to those you served with? |
| 5. How did you begin to establish new friendships/relationships when you left the military? |
| 6. What were the challenges for you in forming new friendships or relationships when discharged? |
| 7. If you were with a partner and/or had family, how did they establish new friendships? |
| 8. How do you feel now about your past and present friendships/connections? |
| 9. Are there any differences between the comradeship you experienced in the military and your civilian connections? |
| 10. What advice would you give a fellow veteran to prevent social isolation and loneliness? |
| 11. How are you managing to stay connected with others during the COVID-19 pandemic? |
| 12. Do you wish to add anything else about your thoughts or feelings on loneliness and social isolation? |

**Fig 1. Semi-structured interview schedule.**

provided a participant information sheet and a consent form. All participants were given the opportunity to ask any questions. If they were still happy to participate, they provided consent and a semi-structured interview was arranged.

Interviews took place between 10th July– 21st September 2020. The interview process was adapted due to COVID-19 and participants were offered interviews by telephone or video call. Of the 11 participants, five opted for telephone and six chose video call. Interviews took place in a private and confidential workspace in the researcher's home and lasted an average of one hour. Prior to the interview, the nature of the research was explained to the participant, and they were reminded that the interview would be recorded. They were also advised that the interview could be paused or terminated at any time, and they could withdraw from the study. Demographic details on age, service type, rank, service length and discharge date were taken from each participant during the interview. At the end of the interview the participant was emailed a participant debrief. Interviews were then transcribed verbatim by the researcher and each transcript was given an ID code. Demographic information was entered onto a spreadsheet that linked the ID code to the transcription to enable the researcher to identify each participant should they wish to withdraw from the study. The coded transcripts, coded spreadsheet and consent forms were all stored separately and securely on a password protected computer.

## Analytical strategy

The interview transcripts were analysed using the principles of reflexive Thematic Analysis [37, 38] as it is a flexible approach to understanding data, is not fixed to a specific epistemological position and "...*can be conducted within both realist/essentialist and constructionist paradigms*..." [38]. Reflexive Thematic Analysis enables the researcher to derive knowledge from the data through an iterative process to identify patterns and meanings within the data that produces codes and ultimately generates final themes [39].

Analysis was conducted by the researcher with consideration for each of the six phases of Braun and Clarke [38] step-by-step guide. Braun and Clarke [37] state their 2006 paper provides a starting point and is not intended to be a rigid procedure that one must adhere to, but rather a fluid approach. They also suggest that researchers should be explicit when writing about theme generation and both points are reflected in the analysis strategy and the analytical trail. Inductive analysis was utilised which ensured the findings were data driven rather than seeking out data that upheld a specific theory. Inductive analysis was utilised to generate data-driven codes, sub-themes, and themes.

As the researcher undertook and transcribed the interviews this enabled them to become fully immersed in the data from the outset. During this phase of analysis, initial impressions were noted e.g. grief, bonding, identity, language. All analysis was done using pen and paper. The transcripts were then read repeatedly so the researcher became familiar with the full dataset and any potential meanings relevant to the research topic were highlighted in red.

Repeatedly reading through the highlighted text generated initial codes which were given separate headings and organised chronologically from discharge, transition, and present day to coincide with the research question. Transcripts were also colour co-ordinated according to the participant as it was anticipated this would ensure the codes generated were taken from a broad base of participants. The full transcripts were then re-read to ensure that all relevant data had been captured. Both authors met repeatedly throughout data analysis to discuss the codes, sub-themes, and themes generated in this study.

Reflexivity is one approach to acknowledging the researcher's position within the research [40]. One author (SGG) was employed by a military charity, and they were acutely aware of

how their experiences may influence the research. It was their intention to remain objective throughout the research process, therefore, they kept a reflexive journal to minimise their biases. This was supported by regular meetings with GWM to discuss the research aims, data collection tools, and data analysis/interpretation. It is noteworthy to remark how the interview schedule questions may have been influenced by the researcher's positionality. For example, the language used in the questions such as the "social impact" of geographical relocation and "challenges" in forming new friendships imply difficulties were experienced and would have prompted the participant to discuss these.

## Results

Given the inconsistencies between the understanding of loneliness and social isolation, participants were asked the meaning of the terms to ensure the analysis was accurately reflected. Participants widely defined loneliness as "*. . .the absence of meaningful relationships. . .*" *(Participant 04)* and suggested it "*. . .isn't necessarily being on your own.*" *(Participant 03)* as you can be "*. . .lonely in a crowded room. . .*" *(Participant 04)*. Social isolation was defined by participants as a lack of physical contacts by "*. . .literally having nobody around you. . .*" *(Participant 04)* and "*. . .where you are on your own.*" *(Participant 09)*.

Three themes were generated from this data: a sense of loss; difficulty connecting in civilian life; and seeking out familiarity (Table 2). Each theme includes sub-themes.

### A sense of loss

**Formation of bonds.**   In order to comprehend the sense of loss, it is important to highlight the strength of military comradeship experienced by the participants. Participants described how strong attachments developed due to being together in prolonged close proximity during their military service, where "*. . .you make very fast, very firm friends for that period of time*" *(Participant 11)*. This was felt to contribute to the unique formation of bonds that operate at a profound level. Having been through the same experiences was integral to shaping friendships and strengthening bonds:

> "*You form a bond, you've been through the same thing together and whether you're a sniper or a woodworker or a cook or a butcher, you've all been through the same military training and you've all had the same good times and you've all been through the bad times and been to the same places and I think that does create a bond certainly.*" *(Participant 02)*

Experiencing intense situations where your life depends upon those with whom you serve was felt to increase the intimacy and depth of attachments. Furthermore, loyalty and respect flourish in challenging situations and participants reported their trust in each other, thus breaking down barriers and cementing a strong supportive network through comradeship:

**Table 2. Themes and sub-themes generated from the dataset.**

| THEME | SUBTHEME |
|---|---|
| **A Sense of loss** | • Formation of bonds<br>• Loss of identity<br>• Detached social networks |
| **Difficulty connecting in civilian life** | • Different social norms<br>• Experiential differences |
| **Seeking out familiarity** | • Reconnecting to ex-military community<br>• Connecting through shared interests |

*"...you have, no matter what, the loyalty is massive, you know what I* mean?*" (Participant 01)*

*"...you've got to have that respect because you're going to a war zone..." (Participant 09)*

*"...having lived through adversity with the people you can trust, you have a high opinion of them..." (Participant 10)*

*"I think one of the things that stick with me is comradeship and all the rest of it is when you've been thrown into an intense situation and it's how...how barriers* break down.*" (Participant 03)*. Having a common goal was felt to bind people together and form a collective unit that operates as a supportive network. Participants reported how bonding is achieved through that supportive network and "fighting" for one another:

*"They say people don't fight for their country they fight for their pals and I think that's right really. People, you know, they support each other often in very difficult situations." (Participant 07)*

The degree to which bonds were developed was evident through the use of phrases such as "*band of brothers*" *(Participant 01)*, illustrating a tremendous sense of kinship experienced within military service. The strength of relationships created within the military was evidenced by the nature in which participants associated it to family. Forming a strong familial attachment is a significant aspect of military life which provides deep and fulfilling relationships:

*"So from my perspective, you know, it was everything, it was family, you know, it was a whole life...yeah, they were family, mates I served with, they were brothers." (Participant 10)*

Although female participants did not use the phrase "band of sisters" they still described their relationships in the military as "family":

*"When I was serving it was excellent, it's like you almost built up another family.... everybody was very supportive and it was like having your own family out there which was really, really good." (Participant 09).*

**Loss of identity.**    Participants described leaving the military as a loss of identity and was described as being similar to feelings of bereavement:

*"When I first left, I was totally and utterly bereft. The day I handed my ID card in I sat and wept because that was my whole identity..." (Participant 05)*

*"I suppose there's almost a bereavement there, having left the forces. Yeah...it's that sense of loss." (Participant 07)*

Following discharge, the dichotomy of military and civilian life felt like living two lives, but comfort was achieved through their ongoing military identity:

*"I often think when I'm out and about I live two lives...I live my life and I live the life that people see. In my head I still lead a military life." (Participant 08)*

*"...the military side I'm more myself than my civilian side." (Participant 05).*

For some, this sense of grief and loss of identity was felt later in life:

*"...it hit me years and years later whereas I know it hits some people instantly and they don't adjust...they find it a real struggle and for me it wasn't, but certain aspects of it will come back as time has gone on." (Participant 02)*

This highlights the importance of military identity and the distinction between transient and chronic loneliness. The impact can be felt immediately upon discharge or many years later, which can inform the positioning of interventions to combat loneliness and social isolation.

**Detached social networks.** No longer having a network of close, like-minded friends felt both lonely and isolated once discharged. The strong attachments that had formed whilst serving were felt to be broken which may result in feelings of social isolation without a supportive network:

*"...coming out the military right, you expect everybody...you expect to keep that connection and it doesn't happen and it's quite a big, big thing because you've been where everybody was around you, you kept in touch and everybody says they'll keep in touch and when you get out it's as if you're just forgotten about." (Participant 09)*

Losing contact with the 'military family' also contributed to feeling lonely and abandoned without anyone to depend upon:

*"I think people struggling with mental health anyway...I think for them to feel so abandoned, that is what worsens everything for them." (Participant* 09)

*"...it's an enormous wrench leaving that family that is the army that is the regiment, the company, the squadron whatever it might be. They're dumped in the big wide world and it's a pretty soulless place really." (Participant 07)*

Missing the social life that the military provided was evidenced and for some the sudden lack of support network on discharge had not been considered:

*"I still really, really miss the social life from the Sergeant's mess, I really, really miss that...you know." (Participant 08)*

*"I hadn't planned that I would have missed it, but I clearly did and I had lost that support network and having all those people around." (Participant 10)*

Those trying to reconnect to childhood friendships found *"...they're just not interested so I felt very isolated when I first came back here" (Participant 05)* and could result in *"...feeling like a stranger in your own town..."* (*Participant* 01). There was some regret leaving the area where they had served as it was a struggle reconnecting: *"If I'd have known then what I know now, I wouldn't have left [area] because that is where I was stationed when my service came to an end." (Participant 08)*. However, not all participants felt this way as one participant had moved back to their childhood home and *"...reconnected with those friends from school and college" (Participant 04)*. Another who had settled close to where they had last been stationed had already forged civilian relationships in the area *"I suppose I had some contacts up here already and I just built on those..." (Participant* 03).

## Difficulty connecting in civilian life

**Different social norms.**   Entering into a new civilian environment can be difficult and the disparity between the military and civilian social norms was defined as "...*a big culture shock*" *(Participant 07)*. In particular, language, humour, and speaking frankly felt very different in civilian life which resulted in feeling like "...*a fish out of water*" *(Participant 08)*.

Participants described the differences in military and non-military language and felt that they were sometimes misunderstood. Differences in what is acceptable language could result in being guarded about what is said:

"...*I almost felt that when I first got out, I couldn't speak freely I had to watch everything that I said. So that kind of silences you a little bit.*" *(Participant 09)*

Humour was a factor that could create a barrier to friendships between ex-military personnel and civilians which resulted in having to adapt interactions:

"...*you can tell the story that you know a military audience will find that funny but it might be too risky to tell that in front of a different audience in a different social situation.*" *(Participant 06)*

Speaking open and honestly could result in fewer social interactions causing isolation:

"*I wasn't prepared to be spoken down to and I would challenge and that would put people's backs up and that would distance them from me...*" *(Participant 08)*

One participant who had worked with civilians whilst serving did not experience the profound disconnect and felt equipped to deal with civilian social norms:

"*I had a foot in both camps so I could work with military people and I could work with civilians, so I was fortunate there.*" *(Participant 07)*

**Experiential differences.**   Participants felt some difficulty connecting with civilians as they did not share the same experiences:

"...*they don't see what's inside me, they don't see what I've experienced, so they don't see, they don't give me credit to be able to do things.*" *(Participant 08)*

What did provide solace was speaking with ex-military personnel as there is a mutual understanding that requires no clarification:

"...*the comfort for me is the fact that you don't have to explain yourself. You don't have to necessarily explain your reaction to certain things.*" *(Participant 11)*

Not being understood or able to share personal experiences made it difficult to open up to civilians:

"...*having to park it even though you didn't want to, you know, having had a sort of overwhelming experience and then not being able to share it as it were, or to have other people understand it and even now talking about it makes me feel slightly emotional and actually difficult.*" *(Participant 06)*

This made individuals feel lonely as they were unable to express their feelings to civilians as they did not have that shared sense of understanding:

"*I was never alone, I was never isolated without anyone, but I could feel lonely in a crowded room. I could have felt absolutely lonely because at the time there was no one that got me, nobody who understood me and therefore nobody I could talk to about what, you know, things that were playing on my mind. . .*" (Participant 11)

Ex-military personnel may not have confidence in civilians to understand their reality and expect them to react negatively if they hear them speak in a way that was acceptable in the military. Due to a lack of trust in civilians, they may remain silent around them:

"*. . .I suppose I have issues with. . .I don't know. . .trust. It's almost like I have lots of friends that I go out with but not who I would say that I would 100% trust*" (Participant 09)

### Seeking out familiarity

**Reconnecting to ex-military community.**   Feeling disconnected in civilian life led to seeking out familiarity via regimental unions where connections were already established:

"*I tend to go to reunions. . .you know. . .regimental unions and I always come away from that feeling so much better, you know, a foot taller because I've seen me mates who've just had a crack on, a laugh.*" (Participant 01)

Meeting up with comrades felt like "*. . .fitting into a comfy pair of slippers. . .*" (Participant 11) and seeing those who were well known brought about a certain level of ease as it presented harmonious interpersonal transactions:

"*It may have been 40 years ago for a lot of these guys, but when they get together in a room within five minutes everyone's playing exactly the same roles they played 40 years ago. It's like they've never been apart and then they'll go back to their own lives.*" (Participant 02)

There was an affinity connecting to ex-military in a civilian setting as there was a mutual understanding of past experiences, and identifying other ex-military in civilian life provided an element of solace:

"*. . .the senior partner there was I suppose the oldest partner in the firm, he'd done national service as a second lieutenant. . .so he had an empathy from where I'd come from.*" (Participant 07)

"*On a night out I can spot, men more than women, but I can spot service personnel I can almost spot from a distance, ones that are still serving and ex ones*" (Participant 11)

Being identified as being ex-military offered common ground:

"*. . .I'll be asked "you're ex-military, aren't you?" and I go "how do you know that?" and they say they can just tell and I think they maybe say things and I say I get it and they say they know you'll get it.*" (Participant 09)

Moving to an area where there was a large ex-military community provided familiarity where connections could easily be established which highlights the importance of geographical location when seeking out others who have shared experiences:

"*I would say here where I live, a village [name] as you can imagine, it has a very big ex-forces community.*" (Participant 06)

Although there was strong evidence of wanting to connect to ex-military personnel, some participants wished to move on from their military career, completely immerse themselves into civilian life and were keen to find alternative ways of connecting with others:

"*. . .you can't replicate that life and really I don't think it's particularly healthy trying to as well.*" (Participant 10)

**Connecting through shared interests.** Searching for others who had similar interests were found through hobbies which provided the opportunity to create civilian networks:

"*Everything from bowling, line dancing, craft, indoor bowling and all sorts of things and through that I then made lots of local pals of which I'm very grateful for cos I had no old friends.*" (Participant 05)

Hobbies were also comparable to military service in that it provided a positive community:

"*. . .in some ways the outdoor community is not dissimilar to the military in many ways. It's quite a positive community, does that make sense? I say that from going back to how many years since I was in the [service type] that it was a positive community.*" (Participant 03)

Support groups provided close connections, but there remained a gravitational force towards ex-military personnel:

"*. . ..it was through a [name] support group. . .within that were other ex-military, believe it or not, and they are my closest friends, not people I met in the military, but people after it. . .a lot of those friendships are my civvy friends, but also happen to be ex-military.*" (Participant 11)

## Discussion

This study aimed to examine the unique factors of loneliness and social isolation for ex-military population from discharge, through transition, to the present day. Findings provide a rich understanding of the risk factors for loneliness and social isolation in this subpopulation of military veterans, specifically the sense of comradeship during their military career and the subsequent sense of loss when leaving the military, difficulty connecting to civilians, and seeking out familiarity with other military veterans. This data can be viewed through the lenses of both the Social Needs Approach and Cognitive Discrepancy Model theories [31, 32].

Comradeship and the intensity of forming bonds are significant factors in service [19] and this study revealed how social networks and intimate attachments are formed through shared social norms, experiences including active service, and consistency of connection for a prolonged period [29]. Edelmann (2018) coins the term "Culturally Meaningful Networks", a theoretical and methodological perspective focusing on transition from military to civilian life in the United Kingdom [29]. Part of this perspective considers the importance of "*concrete*

*relationships with specific others*" and "*the meaning the actors invest in these relationships by which they enact and understand related interactions*" (Edelmann, 2018, pg. 333). Within the current study there was evidence of this through the strong feeling of 'family' bonds whilst serving, with participants considering their connections as a supportive family where trust, respect and integrity are rapidly formed through their active duty. Furthermore, the Social Needs Approach identifies the absence of an intimate attachment as emotional loneliness and a lack of social network as social loneliness, whereas the Cognitive Discrepancy Model suggests that loneliness occurs when there is a perceived discrepancy between the actual and desired level of social involvement. The emotional and social constructs of the Social Needs Approach and the desired level of social contact of the Cognitive Discrepancy Model are satisfied through the vast network of social contacts and meaningful friendships that are created in-service, and comradeship and the military family act as protective factors against the adverse effects of loneliness and social isolation. However, on discharge, these factors are absent as the supportive network is severed and the quality of friendships suffers. This is supportive of other literature within this subpopulation where ex-military personnel experience loneliness due to reduced support networks [21], have weaker social ties [25] and feel detached from military and civilian society [23]. Given the strength of comradeship and military identity it is important to note that loneliness and/or social isolation may be experienced later in life. A sense of belonging features as a distinct need throughout the study and it is noteworthy that 'thwarted belongingness" is an element of the Interpersonal-Psychological Theory of Suicide [41]. Given the concerns around suicide two years following discharge in young male early service leavers [42], it is vital that a sense of belonging is fulfilled.

Through this sense of comradeship during an individual's military career, 'a sense of loss' was experienced after transition out of the military by no longer being surrounded by a large social network, and a sudden lack of rewarding relationships left an emptiness in some participants' lives. In this study, feelings of loss were mitigated by being occupied with family life or employment, although participants who said they had never felt lonely or socially isolated acknowledged, to some degree, that they missed the close-knit camaraderie of the military later in life. This highlights the complexities of loneliness and social isolation, the mediating pathways as well as the significance of transient loneliness that can be experienced intermittently throughout the lifespan. Both the emotional and social loneliness constructs of the Social Needs Approach go some way to explaining this. Emotional loneliness is the absence of an intimate attachment which is reflected in the data where exiting the military was described as a grieving process with no sense identity or belonging. Transition has been shown to cause identity conflict [17] and searching for continuity between military and civilian life can make it difficult to adjust [15] as this study demonstrates. On the other hand, social loneliness is characterised by a lack of social network [31], and this was identified through 'Detached Social Networks' as social connections were severed on discharge. Social ties are weaker during transition [25] and reduced social networks increases loneliness [21] which was identified in this study as well as a sense of abandonment without the supportive network to rely upon. Reconnecting to childhood friendships rarely worked out, resulting in a detachment from the hometown, however, those who had settled close to where they had been last stationed were able to bridge this gap as they had already formed connections prior to discharge. The findings can also be related to the Cognitive Discrepancy Model as there was a deficiency in the quality and quantity of relationships: participants lost meaningful friendships and no longer had access to their vast social network. Conversely, those who had already established quality relationships and built up a social network prior to discharge were less likely to feel lonely and/or socially isolated.

During transition, "reverse culture shock" may be experienced [16] and this study highlights this as participants described their transitional experiences as a culture shock and having lived two different lives. This change in culture led to feelings of detachment from both military and civilian society [23] and being unable to relate to civilians [24] due to military social norms [12]. Involvement with the Royal British Legion has helped reduce the impact of these feelings by offering social support akin to comradeship and where ex-military have been able to construct a "modified military self" during the transitional period of identity challenge [43]. The subtheme 'Different Social Norms' revealed the differences between military and civilian cultures and the difficulties they present during transition [29]. The disparity between military and civilian social norms could make life very lonely as, despite being surrounded by others, it was difficult to communicate or be understood which resulted in trust issues, feeling silenced or having to adapt language, akin to experiences of emotional loneliness (Social Needs Approach), and is reflective of the Cognitive Discrepancy Model where loneliness and social isolation are caused by a dearth in the quality of meaningful relationships. Interestingly, this divide was bridged for those who worked alongside civilians whilst serving as they were already aware of and prepared for the cultural differences.

Losing the 'military family' and no longer having a reliable social network provided the motivation to seek out fulfilling relationships where relationships could develop both socially and emotionally. Peer support networks provide a shared sense of social identity and emotional support [44]. For some, there is a desire to reconnect to the military community through reunions where established relationships provide harmonious interpersonal transactions and promote well-being as there is mutual understanding. Crucially, these sought interactions are not quantifiable but are meaningful. Others may seek out social connections via hobbies or groups where interests are shared, and meaningful relationships develop. A positive transition is associated with being connected [12] and this study highlights that loneliness and social isolation are central to transition. Whether it is reunions, hobbies, or both that are harnessed to prevent loneliness and social isolation, what is clear is that they promote positive social interactions where common beliefs and goals are shared and a sense of belonging, similar to the experiences of camaraderie, is achieved. These pathways satisfy the three factors of Edelmann's "Culturally Meaningful Networks" structure, meaning and time [29] in preventing loneliness and social isolation following military service. It can take time to connect during transition, but there was evidence to suggest that those who worked alongside civilians whilst serving or remained close to their geographical station on discharge enabled them to bridge the gap between military and civilian life, hence a smoother transition and less likely to be lonely or socially isolated. Interestingly, male and female participants both described military connections similarly and no gender-based differences were apparent from this data, although this was not a focus of the current paper.

## Consideration of interventions

This study has shown that there is unlikely to be a suitable universal approach where a 'one-size-fits-all' intervention will diminish the effects of loneliness and social isolation in the ex-military population. Successful interventions have yielded a reduction experienced loneliness by connecting ex-military personnel via telephone [45], reduced social isolation through volunteering [46], socially reconnected through various outdoor horticultural activities [47] and maintained well-being through a community-based peer support programme [44]. Charitable organisations aim to tackle loneliness and social isolation through telephone-based interventions such as the Royal British Legion [20] Branch Community Support and SSAFA's Forces line [48]. The history and heritage projects via Soldier On! [49] and residential courses that

encompass outdoor activities such as Future4Heroes [50] and The Warrior Programme [51], all assist with enhancing social interaction between veterans.

Recent guidance developed by the Campaign to End Loneliness suggests that interventions reduce loneliness either by supporting existing relationships, helping people to make new connections, or allowing individuals to change their thinking about their social connections [5]. It is suggested that mixed (veteran and civilian) social groups may support veterans to change the way they consider their social connections, and to consider non-military friendships. An advantage of this would also enable civilians to learn about military culture and perhaps the onus should not merely lie with the ex-military community but also the broader population to alleviate the disconnect and promote inclusivity. The benefits of ex-military integrating with wider community services has been evidenced in a recently undertaken Delphi study [30].

Differences between participants suggest the positioning of interventions needs to be considered. Some felt lonely and socially isolated in-service, whereas others located near to their last station or had worked with civilians whilst serving which assisted with a smoother transition as they were able to connect with ease in civilian life. This indicates that interventions prior to discharge should be considered. Participants could experience loneliness and/or socially isolation immediately on exiting the military, whereas others experienced this years later signifying interventions should be tailored to meet individual needs and available from discharge, during transition and throughout the lifespan. What is of paramount importance is having a sense of belonging where individuals can socially connect, share experiences, and allow meaningful relationships to flourish.

Finally, it is important to consider the impact of the COVID-19 pandemic on interventions in this field. Many of these projects, however, have had to adapt due to the COVID-19 pandemic and offer alternative online support at present. Loneliness interventions have had to rapidly shift their focus away from face-to-face programmes to meet COVID-19 guidance, often resulting adaptation and remote delivery [5].

## Strengths and limitations

The theoretical underpinnings of this study of loneliness and social isolation in the British Armed Forces ex-military population is a key strength. A further strength is the broad range of perspectives from males and females from the Royal Navy, Army, and RAF, as well as ranking from Private to Lieutenant Colonel. However, a limitation of this study may be the under-reporting feelings of loneliness and social isolation as military cultural norms may prevent emotional disclosure [52]. For example, a notable difference in this study was that females were more likely than males to acknowledge feeling lonely or social isolated which may be due to the propensity of males under reporting feelings of loneliness and social isolation [53]. Time elapsed since discharge ranged from three to 38 years (mean = 20 years) and results may be subject to recall bias [54]. The length of time since active service, or current employment status, was not considered as part of this study and should also be addressed as a limitation. These factors may have had an impact on the individuals' experiences of loneliness and social isolation and should be addressed in future work. None were early service leavers, and this cohort are particularly at risk [55]. Although preconceptions were minimised by reflexivity, the language used in some of the interview questions implied difficulties with socially connecting which may have prompted participants to discuss their challenges.

## Future directions

It would be prudent to obtain further accounts from a wider range of ex-military personnel on loneliness and social isolation to capture a broader understanding of experiences. There was

little evidence of differences between rank and service type and no evidence of differences in duration of service and further research should consider looking into this further. It would be worthwhile conducting longitudinal research and include early service leavers and those who have been medically discharged who are plunged unexpectedly into civilian life. Finally, research should also include an examination of current interventions that tackle loneliness and social isolation, their effectiveness and how they benefit the ex-military population.

## Conclusion

This study highlights the complexities of loneliness and social isolation and their unique impact on discharge and transition, and throughout the lifespan. The Social Needs Approach [31] and Cognitive Discrepancy Model [32] support insight into veterans' experiences of loneliness. The military enabled individuals to develop quality bonds through shared meaning and experience that not only reduced feelings of loneliness but also social isolation through close and prolonged contact. The sense of belonging was key to social connection but was severed upon discharge through lack of consistent connection. Feelings of dissimilarity with the civilian population, through dissimilar social norms and a lack of connection, hindered new social connections and led individuals to seek connections with like-minded military veterans. It is essential that further qualitative research is conducted to establish a broader range of British Armed Forces ex-military personnel perspectives. This will help inform best policy and practice and appropriate intervention strategies to minimise the effects of loneliness and social isolation for ex-military personnel and future service leavers.

## Acknowledgments

The authors would like to express thanks to the eleven participants who were willing to give up their time to share their experiences openly and honestly.

## Author Contributions

**Conceptualization:** Suzanne Guthrie-Gower, Gemma Wilson-Menzfeld.

**Data curation:** Suzanne Guthrie-Gower.

**Formal analysis:** Suzanne Guthrie-Gower.

**Methodology:** Suzanne Guthrie-Gower.

**Supervision:** Gemma Wilson-Menzfeld.

**Writing – original draft:** Suzanne Guthrie-Gower.

**Writing – review & editing:** Suzanne Guthrie-Gower, Gemma Wilson-Menzfeld.

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
