## [Decision Letter · Decision Letter 0]

2 Feb 2022

PONE-D-21-28325Ex-military personnel’s experiences of loneliness and social isolation from discharge, through transition, to the present dayPLOS ONE

Dear Dr. Wilson-Menzfeld,

Thank you for submitting your manuscript to PLOS ONE. After careful consideration, we feel that it has merit but does not fully meet PLOS ONE’s publication criteria as it currently stands. Therefore, we invite you to submit a revised version of the manuscript that addresses the points raised during the review process.

Please see my comments below alongside the reviewers and respond to both.  Please submit your revised manuscript by Mar 19 2022 11:59PM. If you will need more time than this to complete your revisions, please reply to this message or contact the journal office at plosone@plos.org. Please include the following items when submitting your revised manuscript:A rebuttal letter that responds to each point raised by the academic editor and reviewer(s). You should upload this letter as a separate file labeled 'Response to Reviewers'.A marked-up copy of your manuscript that highlights changes made to the original version. You should upload this as a separate file labeled 'Revised Manuscript with Track Changes'.An unmarked version of your revised paper without tracked changes. You should upload this as a separate file labeled 'Manuscript'.

We look forward to receiving your revised manuscript.

Kind regards,

Andrew Soundy

Academic Editor

PLOS ONE

Journal Requirements:

2. Please note that in order to use the direct billing option the corresponding author must be affiliated with the chosen institute. Please either amend your manuscript to change the affiliation or corresponding author, or email us at plosone@plos.org with a request to remove this option.

Additional Editor Comments:

comments on methods

Please identify the type of phenomenological approach that was adopted

Please situate this with a paradigm you mention in the analysis section about a realist/essentialist and constructionist. – please up front select one and identify the ontological position of this and the epistemological position. Think about articles like this to consider the point https://www.tandfonline.com/doi/abs/10.1080/2159676X.2017.1393221

also may be use wording from the authors website which identifies that there are different ways approaching thematic analysis https://www.thematicanalysis.net/understanding-ta/

You need to consider a framework like Obrien et al (2014) the SRQR or the COREQ

line 96 to 100 you identify demographics (results) for your participants – can this be moved.

You identify a sampling strategy but not a rationale for sample size this is needed.

Re the interview guide was it piloted or did you undertake a cognitive interview? Please also identify if there were areas or domains you wanted to cover and how they were selected.

In the analysis – tell the reader less about what it is gernally and more about the how you did it e.g., as above you say an inductive approach but Braun and Clarke link this within a paradigmatic view https://www.thematicanalysis.net/understanding-ta/

In an appendix please add an audit trail – examples of each of the 6 stages please so the reader can see the how.

Add in a section on trustworthiness or quality for the reader

Reviewers' comments:

Reviewer's Responses to Questions

**Comments to the Author**

1. Is the manuscript technically sound, and do the data support the conclusions?

Reviewer #1: Partly

2. Has the statistical analysis been performed appropriately and rigorously? 

Reviewer #1: N/A

3. Have the authors made all data underlying the findings in their manuscript fully available?

Reviewer #1: No

4. Is the manuscript presented in an intelligible fashion and written in standard English?

Reviewer #1: Yes

5. Review Comments to the Author

Reviewer #1: Manuscript Review

Journal: PLOS ONE

Title: ‘Ex-military personnel’s experiences of loneliness and social isolation from discharge, through transition, to the present day’

Overview: The manuscript focusses on the loneliness and isolation of former military members in the workplace. More specifically, the researchers’ objective is to shed light on the experience of the transition from military to civilian life.

Strengths: The abstract is engaging, and the article reads well; the quality of the writing flows and is mostly free of typographical errors. In addition, the paper adopted some organisational strategies to present the data which helps the reader. Specifically, the authors prepare the reader for what is to come using the standard divisions for the manuscript effectively, and the presentation of the results using an orienting table and a generous use of quotes, which are very positive aspects. The bold headings also added to the overall clarity, flow, and organisation of the paper. In the result section, quotations are contextualised concisely to weave an engaging presentation. Finally, the study is used to describe some phenomena – loneliness and social isolation – and the methodology chosen makes that possible (although the rational for that choice could have been better explicated – see below). Theories are integrated in the manuscript introduction and revisited in the discussion; therefore, their introduction at the start of the manuscript, and a return to them in the discussion also helps the paper flow and does not leave the reader searching for a conclusion. There are clear take-home messages that are communicated.

Weaknesses: There are four substantively problematic issues with this manuscript:

1. The first one is the limited review of some of the literature at the international level, given the sizable presence of relevant publications on veterans and transitions from the military life which would inform this presentation. Also, where much of the review appears (in the discussion) is not as effective a strategy to inform the reader of the gap the study is aiming to fill than if studies were introduced right from the start. Returning to the review of key studies in the discussion to contextualise the results and demonstrate how the findings met the objective of the research and how they add to the literature, would work significantly better. (Although page 3, lines 80-86 provides an initial attempt to do so, introducing new literature in the discussion rather than in the introduction undermines the authors’ presentation strategy.

2. Given that this is a qualitative study, regardless of having chosen phenomenology as the methodology, it would be helpful to have a section on reflexivity in the paper so that the reader can situate the perspective (insider/outsider a.k.a., their epistemological position) of the authors.

3. The absence of a critical discussion regarding the military culture is a missed opportunity to permit a much better and deeper understanding of the factors that make this transition from the military so difficult (e.g., how the military gains by maintaining and reinforcing the attitude and belief amongst its members that no one outside the military can understand the military experience). Several opportunities to gain insights in this regard seemed to have been missed (e.g., page 15, lines 333-334: is there a boundary condition dividing those who would like to remain in the culture of the military and those who wish to leave it).

4. Most importantly, the ‘slippage and leaps’ present between the quotes and their interpretation (detailed below) is a concern that must be adressed.

Introduction:

As mentioned above, a more inclusive international review is missing here and some of the literature reviewed in the discussion could be moved to the introduction. The most pertinent references can then be revisited in the discussion after having presented the results.

Although the use of acronyms is common, limited use is always appreciated by readers. In this manuscript, the acronyms for Social Needs Approach (SNA) and the Cognitive Discrepancy Model (CDM) do not appear that frequently. Therefore, I would recommend writing the terms out as

spelling them out will bring clarity for those not familiar with this nomenclature. I would also add a definition of the ‘third sector’ to increase the accessibility of the manuscript for individuals outside the UK, non-profit sector or NGOs.

Design:

(Lines 90-93): This was a very brief presentation of the research approach chosen. Perhaps it could be expanded to include a brief description and explanation of the phenomenological approach, provide a couple of references, and explain why it was chosen instead of other qualitative methodologies (e.g., Discourse analysis, Grounded Theory). Moreover, in the social sciences, not everyone, and especially not those new to qualitative methodologies, necessarily mean the same when they use the term ‘Phenomenology’ or a ‘Phenomenological approach’. (Line 144) This is especially true for students and new scholars with a scientific historical background. Especially for their benefit, the authors could include a brief, concise definition of inductive research, and again, provide a couple of references.

(Please note that on line: 93, the university name was made anonymous, yet, it is included in the accompanying information on PDF page 3.)

Participants:

(Page 4, Lines 96 to 100) This section starts with a major challenge, and one which impacts many other aspects of the manuscript; namely, the fact that there are only 11 participants. Why 11 participants? Was thematic saturation reached with 11 participants? The authors do not explain the reasons for this number of participants. There also should be some background information provided on the number of veterans in the UK and a breakdown of their age, distribution across the three divisions (Navy, Army, Air Force), their gender, past and current marital status, rural or urban living environments, etc. with which to compare the present sample.

Confidentiality and privacy are legitimate concerns in qualitative research to limit the availability of some information, especially when considering how to present demographic information to maintain anonymity. Thus, the demographic information that was provided was much appreciated; however, some information remains essential. In order to determine the potential impact of certain key factors, some details must be presented, albeit carefully. These will help the reader to understand, contextualise, and interpret the results. For example, what was the present employment status of these participants? Were all participants completely retired? Did some have post-military careers? At the time of the interview, how long had they been out of the military? A table should be provided indicating this information. Also, some of this information is presented in the discussion (page 22, lines 502-507) but this belongs to the participants’ section so that the reader can read the results with this information to better understand what potential impact they may have on the data. Therefore, minimally, this agglomerated information should first be presented in the participants’ section; not in the discussion. Without it, it is very difficult to get a sense of the value of the findings. This will become more evident further into my review, given some of the conclusions put forward, for example, about gender.

Materials:

I very much appreciated the inclusion of the interview guide. However, the questions suggested the direction the researchers wanted the participants to answer. (e.g., ‘If you moved to a different area, how did this impact you socially?’ or ‘What were the challenges for you in forming new friendships or relationships when discharged?’ if one suggests that there were challenges, the participants are bound to speak of challenges, whereas if the question had been ‘Can you tell me about your social life when you were discharged? Or even better, can you describe your transition from the military?’ One would gain insights regarding what was salient in the experience of the participants. Prompts can always be introduced later but with a more neutral approach: ‘On a day-to-day basis, what about work-wise/Socially/Psychologically’ and ‘What was your daily routine like when you left the military?’ Etc. This way, participants can speak of their experience, rather than be taken to a place where the researcher decides what is important for participants to discuss.) Although this aspect cannot be changed as the interviews are already conducted, it is a significant limitation that should be acknowledged by the authors and the related implications should be discussed in terms of weaknesses.

Procedure:

When were the interviews taking place? (Aside from during COVID) from what date to what date? What was the professional network chosen to circulate the description of the study and recruit participants? Why was only this network chosen for the recruitment? It is interesting that on page 5, lines 119–122, Five participants chose to take part in an interview over the telephone and six preferred a video call interview. Were there differences observed between the two ‘groups’, any boundary condition separating these two ‘groups’)?

Analysis Strategy (consider the following heading instead: Analytical Strategy)

(Page 6, line 134) It would have been helpful to know how the analysis was carried out. For example, did the researchers use a software program such as NVivo to aid in their reflexive thematic analysis? Or another specialised program? Given the style of referencing which uses numbers for referenced articles, also using a number system to refer to participants can be somewhat confusing, especially that there is some overlaps. For instance, in one sentence, the authors state that “Of the military literature, only (21) consider loneliness and social isolation from a theoretical perspective.” Understandably, with a small sample, limited descriptive information can be attached to each specific participant in order to maintain anonymity; however, using discontinuous pseudonyms might help with clarity and further protect the participants’ identity. (Additionally, in some places, mentioning the names of the authors of the references might help (e.g., page 3, lines 69-70). (Page 6, line 136) There is a quote without a clear reference. One would assume that it is either referring to the reference number 32 or 33 mentioned earlier in the sentence but which one is it and the page number is missing.

Results:

As mentioned earlier, the quotes are contextualised concisely. However, some of the preambles make a leap between what is said in the quote and its presentation/interpretation. Ffor example, on page 8, lines 181-183, the interpretive preamble speaks of ‘trust, reliability, and respect’ and ‘allegiance’ but the quote actually speak of ‘loyalty’ and ‘comradeship’, which are actually not the same. Therefore, although these sentiments might be related at times, there is slippage between the quotes and the meaning given to the quote). The entire result section should be carefully reviewed, and the misalignment between the quotes and their interpretation should be corrected either by changing or adding a quote, or being careful not to misattribute meanings that are not in the data.

(Page 9, line 193) The introduction of the phrase ‘band of brothers’ to describe a sense of kingship between soldiers is an opportunity for some critical examination of the culture of the military. Yet, it is surprising that this did not trigger a gender-based analysis of the data. This is not problematised in terms of how women, in comparison with men, might feel when transitioning from the military. Extensive research about women in the military and women as veterans in various countries have documented how women have experienced this transition very differently from men. Women have dealt with harassment and sexual violence from male colleagues, they have had to contend with these behaviours which have been linked to the military culture. Therefore, how does the present data (e.g., page 20, lines 460-463?) fits with this generalised observation? How can this be explained)? This quote alone deserved some unpacking (e.g., did the expression a ‘band of sisters’ come up? Why so or why not? What is the power dynamics in this situation? Given the increasingly extensive literature on this topic, not examining this dynamic with more critical depth is a gap that should be addressed.

(Page 9, lines 199-200) The following sentence by the researchers “The military family is comparable to the parental instinct, or siblings looking out for one another which provides a unique level of intimacy where unconscious intersubjectivity is experienced” is not easily understood. Maybe rephrase and expand. Also, in the same section and in the discussion (page 18, lines 410-411), the switch from verbal to non-verbal communication as ‘indicating an even closer bond’ should be more clearly introduced and presented, and this affirmation should also be supported by empirical evidence and references.

(Page 9, line 207) The themes explored are generally well chosen given the focus of the paper. However, I wonder about the label ‘bereavement’ for one of the themes. The term is quite metaphorical in this context (i.e., it is not a literal bereavement associated with death but instead with the loss of an identity). I understand the metaphor, but the title primes the reader for a discussion of an actual death only to realise that it focusses on the loss of identity. Maybe simply speaking of the ‘loss of identity’ or just ‘loss’ would be preferable. Especially since bereavement is mentioned in reference to the Social Needs Approach later on in the discussion (lines 445-447) and in that particular instance, it probably means bereavement in the traditional sense (i.e. related to an actual death), replacing it by the loss of an identity seems more appropriate.

(Page 10, lines 223-227) For the next theme, the preamble to a quote, once again, makes too much of a leap between it and its interpretation. More care has to go into staying true to the participants’ words and meanings, and assuring that the interpretation is well supported by the quote. Possibly, the researchers have more of the interview to draw from to make certain interpretations, but from the reader’s perspective, the quote is the evidence that must convince. Thus, one should avoid slippages and leaps between the quotes and their interpretations, or simply use a different, more fitting quote to permit making/illustrating a particular point. For example, a participant saying you ‘just get on with it’ does not support the authors suggesting that there is a ‘military culture of resilience’ and that it ‘prevailed’. The given quote does not support such an interpretation.

Page 11, lines 239-240 and 242-243 reflect opinions rather than experiences. Given that the study is not an opinion survey but rather a telling of an experience and its meaning, it is not convincing when quotes are chosen of participants speaking of what they ‘think’ rather than relaying what they are presently experiencing or have experienced.

(Page 11, line 258-259) Try to avoid the use of colloquial expressions outside of quotes (e.g., straight talking). Instead use more internationally understood formal language (e.g., speaking in a frank, truthful, honest or direct manner).

The label, ‘speaking a different language’ for this theme was also surprising as it is figurative and not what the quotes in that section are referring to. Maybe ‘different social norms’ could be used instead. There are authors that have spoken about the military use of distancing language (e.g., extensive use of acronyms and jargon), which is what I expected to read about under this theme’s label. However, the quoted interview text in the manuscript appear to allude to social norms rather than using a language others do not understand.

Here again, there is an opportunity for a discussion of how these social norms benefits the military institution, since these are not accidental but are functions/features/results of the military’s culture and total institution’s status (Gofman). So perhaps there is an opportunity to address the root cause by promoting more interaction with civilians and civilian organisations, especially since this was shown in the study to be a factor that seemed to reduce alienation/isolation after leaving the military.

(Page 13, lines 296-298) In the interpretation of ‘Experiential Differences’, the researchers conclude that there is a ‘disconnect between ex-military and civilians in terms of understanding experiences and differences in language’, but the quotes do not support this interpretation. What the quotes suggest is that former military members do not trust civilians to understand their reality and expect them to react negatively if they hear the military members speak in a way that was acceptable in the military (i.e., following certain norms of what is acceptable to say in the context of the military culture). Fearing a negative reaction, they silence themselves.

Typographical errors:

Page 3 line 72, ‘Royal British Legion’ is repeated.

Throughout the manuscript ‘the researcher’ is referred to in the singular. However, on the title page, it says that both authors contributed equally. Therefore, this will need to be edited to be consistent with the authorship of the manuscript.

6. PLOS authors have the option to publish the peer review history of their article (what does this mean?). If published, this will include your full peer review and any attached files.

Reviewer #1: No

---

## [Author Response · Author response to Decision Letter 0]

27 Apr 2022

Thank you to both the editors and the reviewers for considering this manuscript. We have made all changes suggested by the reviewers. We have shown these in tracked changes (unless otherwise stated). We have also included a ‘response to reviewers’ document which documents each revision requested and each change made.

---

## [Editor Report · Decision Letter 1]

26 May 2022

Ex-military personnel's experiences of loneliness and social isolation from discharge, through transition, to the present day

PONE-D-21-28325R1

Dear Dr. Wilson-Menzfeld,

We’re pleased to inform you that your manuscript has been judged scientifically suitable for publication and will be formally accepted for publication once it meets all outstanding technical requirements.

Kind regards,

Andrew Soundy

Academic Editor

PLOS ONE

Additional Editor Comments (optional):

Thank you for addressing concerns and I wish you the best for this important area of research.
---

## [Editor Report · Acceptance letter]

27 May 2022

PONE-D-21-28325R1 

Ex-military personnel’s experiences of loneliness and social isolation from discharge, through transition, to the present day 

Dear Dr. Wilson-Menzfeld:

I'm pleased to inform you that your manuscript has been deemed suitable for publication in PLOS ONE. Congratulations! Your manuscript is now with our production department. 

Kind regards, 

on behalf of

Dr. Andrew Soundy 

Academic Editor

PLOS ONE